# Quantification of Tafenoquine and 5,6-Orthoquinone Tafenoquine by UHPLC-MS/MS in Blood, Plasma, and Urine, and Application to a Pharmacokinetic Study

**DOI:** 10.3390/molecules27238186

**Published:** 2022-11-24

**Authors:** Geoffrey W. Birrell, Karin Van Breda, Bridget Barber, Rebecca Webster, James S. McCarthy, G. Dennis Shanks, Michael D. Edstein

**Affiliations:** 1Drug Evaluation, Australian Defence Force Malaria and Infectious Disease Institute, Brisbane 4051, Australia; 2Clinical Malaria Group, QIMR Berghofer Medical Research Institute, Brisbane 4006, Australia; 3Department of Infectious Diseases, University of Melbourne, Melbourne 3010, Australia

**Keywords:** tafenoquine, 5,6-orthoquinone tafenoquine, analytical method, urine, blood, plasma, malaria

## Abstract

Analytical methods for the quantification of the new 8-aminoquinoline antimalarial tafenoquine (TQ) in human blood, plasma and urine, and the 5,6-orthoquinone tafenoquine metabolite (5,6-OQTQ) in human plasma and urine have been validated. The procedure involved acetonitrile extraction of samples followed by ultra-high-performance liquid chromatography tandem mass spectrometry (UHPLC-MS/MS). Chromatography was performed using a Waters Atlantis T3 column with a gradient of 0.1% formic acid and acetonitrile at a flow rate of 0.5 mL per minute for blood and plasma. Urine analysis was the same but with methanol containing 0.1% formic acid replacing acetonitrile mobile phase. The calibration range for TQ and 5,6-OQTQ in plasma was 1 to 1200 ng/mL, and in urine was 10 to 1000 ng/mL. Blood calibration range for TQ was 1 to 1200 ng/mL. Blood could not be validated for 5,6-OQTQ due to significant signal suppression. The inter-assay precision (coefficient of variation %) was 9.9% for TQ at 1 ng/mL in blood (*n* = 14) and 8.2% for TQ and 7.1% for 5,6-OQTQ at 1 ng/mL in plasma (*n* = 14). For urine, the inter-assay precision was 8.2% for TQ and 6.4% for 5,6-OQTQ at 10 ng/mL (*n* = 14). TQ and 5,6-OQTQ are stable in blood, plasma and urine for at least three months at both −80 °C and −20 °C. Once validated, the analytical methods were applied to samples collected from healthy volunteers who were experimentally infected with *Plasmodium falciparum* to evaluate the blood stage antimalarial activity of TQ and to determine the therapeutic dose estimates for TQ, the full details of which will be published elsewhere. In this study, the measurement of TQ and 5,6-OQTQ concentrations in samples from one of the four cohorts of participants is reported. Interestingly, TQ urine concentrations were proportional to parasite recrudescence times post dosing To our knowledge, this is the first description of a fully validated method for the measurement of TQ and 5,6-OQTQ quantification in urine.

## 1. Introduction

Antimalarial drugs of the 8-aminoquinoline class (pamaquine, primaquine (PQ), and tafenoquine (TQ)) have been the basis of malaria chemotherapy to prevent relapses and block transmission for nearly a century. Despite their wide-spread use, their mechanisms of action and particularly their metabolism have been poorly understood. The potential importance of metabolism was suggested by an extensive 8-aminoquinoline testing program conducted by the US Army which found that 8-aminoquinolines without redox activity were unable to inactivate the latent residual parasites in the liver (hypnozoites) responsible for relapsing malaria [1]. Furthermore, combining these drugs with other antimalarials such as chloroquine has demonstrated marked synergy, and thus these drugs have been frequently used in combinations such as the “CP tablet” of the Vietnam War era (300 mg chloroquine, 45 mg primaquine) [2].

Robust methods of quantification of 8-aminoquinolines and their metabolites are fundamental to improving our understanding of the mechanism of actions of 8-aminoquinolines, and will be essential for evaluating 8-aminoquinoline combination therapies [3]. This is particularly true of the most recently registered (2018) 8-aminoquinoline, TQ, which is available for both weekly chemoprophylaxis and radical cure of vivax malaria when used in combination with chloroquine and in persons known to have normal glucose-6-phosphate dehydrogenase (G6PD) enzyme function [4]. In 1956, it was reported that red blood cells from PQ sensitive persons were deficient in G6PD [5]. Toxicity, particularly from hemolytic reactions in G6PD deficient individuals have defined the tolerance limits for the 8-aminoquinolines [6].

G6PD deficiency is the most common human enzyme defect, being present in more than 400 million people worldwide. The striking similarity between the areas where G6PD deficiency is common and *Plasmodium falciparum* malaria is endemic provides circumstantial evidence that G6PD deficiency confers resistance against malaria [6]. G6PD catalyses the first reaction in the pentose phosphate pathway, in which glucose is converted into the pentose sugars required for glycolysis and for various biosynthetic reactions. The pentose phosphate pathway also provides reducing power in the form of NADPH by the action of G6PD and 6-phosphogluconate dehydrogenase. NADPH serves as an electron donor for many enzymatic reactions, and its production is crucial to the protection of cells from oxidative stress [7]. Apart from the potential for hemolytic reactions, both PQ and TQ have capacity to kill dormant malaria parasites in the liver.

The precise mechanism of action of 8-aminoquinolines such as PQ has been a subject of interest and conjecture for many years. It is known that monoamine oxidase (MAO) and cytochrome P450 2D6 (CYP2D6) convert PQ to the inactive carboxyprimaquine [8]. Using synthetic metabolites, Camarda et al. [9] found that hydroxylated-PQ metabolites (OH-PQm) are responsible for efficacy against liver and sexual transmission stages of *P. falciparum*. The major CYP2D6 metabolite, formed by degradation of the unstable 5-hydroxy-PQ was found to be 5,6-OQPQ [10]. More recently, Fasinu et al. [3] tested the ability of erythrocytes to mediate the formation of reactive oxidative PQ metabolites in the absence of hepatic enzymes. They found that 5,6-OQPQ is produced as a stable metabolite of PQ in human erythrocytes. Reactive oxygen species (ROS) generated by redox-cycling of products in the pathway leading to 5,6-OQPQ have also been proposed to be responsible for the biological activity of PQ. Thus stable 5,6-orthoquinones are proposed to be markers of clinical efficacy.

In a recent study in mice, Vuong et al. [11] found that CYP 2D status influences the metabolic conversion of TQ to 5,6-OQTQ, and that plasma 5,6-OQTQ concentrations were highest in wild-type extensive-metabolizer phenotype mice. They also identified dearylated metabolites in plasma and livers of TQ-treated mice that are precursors to 5,6-OQTQ. Our observation that 5,6-OQTQ is stable in plasma and urine for UHPLC-MS/MS assay validation is consistent with the 5,6-OQTQ being a stable oxidative product of the 5-dearylation pathway. This is different to PQ as the 5 position of PQ is open for direct oxidative modification without dearylation. CYP 2D6 and other cytochromes likely catalyze the conversion of TQ to 5,6-OQTQ in addition to other phenolic metabolites and that one or more of these produce the redox cycling and resultant oxidative stress responsible for TQ radical cure activity [11]. Recently, Pookmanee et al. reported the first validated method for the determination of PQ and 5,6-OQPQ in human plasma and urine [12]; however, we found their method not suitable for TQ-containing samples in the current study. Herein, we describe the methods developed in our laboratory to reliably and reproducibly quantify TQ and its metabolic product 5,6-OQTQ in blood, plasma and urine, and its application in a phase 1b clinical trial. The trial evaluated the blood stage antimalarial activity of single oral doses of TQ in healthy participants experimentally infected with *Plasmodium falciparum* to determine the minimum inhibitory TQ concentration [13]. Participants were randomized to receive differing doses of TQ (200 ng to 600 mg) with the pharmacokinetic results of one cohort (administered 200 mg TQ) presented here. The complete results of the phase 1b clinical trial will be published elsewhere.

## 2. Results

### 2.1. Fragmentation Patterns and MS/MS Spectra of TQ and 5,6-OQTQ

The fragmentation patterns and MS/MS spectra of TQ and 5,6-OQTQ are shown in Figure 1. The MS/MS spectra of TQ (*m*/*z* = 464.2156) showed fragments consistent with the following; *m*/*z* 379.1264 (−86;-C_5_H_11_N), 447.1890 (−17;-NH_3_) and 391.1264 (−58;-C_3_H_8_N); 5,6-OQTQ (*m*/*z* = 304.1656) showed fragments consistent with the following; 287.1390 (−17;-NH_3_), 259.1441 (−45;-CH_3_NO) and 219.0764 (−85;-C_5_H_11_N).

### 2.2. Detection and Quantification of 5,6-Tafenoquine, Stable Isotope Label-Tafenoquine and Tafenoquine

The retention times of 5,6-OQTQ, SIL-TQ and TQ in plasma and blood were 1.6, 2.0 and 2.0 min, respectively. The retention times of 5,6-OQTQ, SIL-TQ and TQ in urine were 1.7, 1.9 and 2.0 min, respectively. Representative chromatograms for each analyte in human plasma and urine are shown in Figure 2.

### 2.3. Method Validation

#### 2.3.1. Selectivity and Specificity

No significant peak interfered with the quantification of TQ and 5,6-OQTQ in the chromatograms of blank human blood, plasma or urine.

#### 2.3.2. Linearity and Sensitivity

The square of the correlation coefficient (r^2^) values were ≥0.99 for all calibration curves. Limits of quantification (LOQ) for TQ and 5,6-OQTQ in plasma was 1 ng/mL, and for TQ in blood was 1 ng/mL. LOQ for both TQ and 5,6-OQTQ in urine were 10 ng/mL. Chromatograms of blank samples with IS, spiked samples at LLOQ, mid quality control (MQC), upper limit of quantification (ULOQ), and clinical trial participant samples for plasma and urine are shown in Figure 2. Linear responses were obtained for TQ and 5,6-OQTQ from 1 ng/mL to 1200 ng/mL in plasma, and for TQ from 1 ng/mL to 1200 ng/mL in blood. Linear responses were obtained for TQ and 5,6-OQTQ in urine from 10 ng/mL to 1000 ng/mL.

#### 2.3.3. Matrix Effect and Recovery

The matrix effect of all analytes was between 86 and 104% according to acceptance criteria [14,15] except for the low quality control (LQC) of 5,6-OQTQ in blood where it was 64 ± 9% (Table 1). This showed significant signal suppression of 5,6-OQTQ in blood. The average matrix effects for TQ and 5,6-OQTQ in urine ranged from 87 to 94% and from 86 to 104%, respectively. The average recoveries in urine ranged from 84 to 96% for TQ and from 91 to 101% for 5,6-OQTQ. The recovery of TQ at LQC (2.5 ng/mL) varied widely from 187% in blood to 69% in plasma. This variation may in part be due to binding of the analytes to plastic surfaces during the recovery analysis and/or due to ion suppression or enhancement from blood and plasma components in spite of matrix effect being low (<3%) for TQ at the LQC level. The %CV for all recoveries was less than 15%.

#### 2.3.4. Accuracy and Precision

The accuracy values of TQ and 5,6-OQTQ in human plasma and blood were within the acceptance criterion of ±20% of LLOQ and ±15% of the QCs. The precision (%CV) values did not exceed 20% of the LLOQ and 15% for each QC as shown in Table 2.

#### 2.3.5. Stability

Both TQ and 5,6-OQTQ were stable in blood, plasma and urine QC standard solutions at 25 °C for four hours and for 24 h at 8 °C (autosampler temperature) with variation of less than 15%. Both TQ and 5,6-OQTQ were stable in blood and plasma at −80 °C for at least 3 months. In urine, TQ and 5,6-OQTQ were stable for four hours at 8 °C and at −80 °C and −20 °C for at least 3 months. Three freeze thaw cycles of either blood, plasma or urine (three cycles of −80 ˚C for 30 min followed by room temperature for 30 min) showed less than 15% variation for both TQ and 5,6-OQTQ at high, mid and low QCs.

### 2.4. Application of the Methods

The methods described here were used to quantify TQ and 5,6-OQTQ concentrations in plasma, blood and urine samples collected from healthy volunteers enrolled in a phase 1b clinical trial evaluating the activity of a single oral dose of TQ against blood stage asexual *P. falciparum* [13]. For the present study, TQ and 5,6-OQTQ were measured in samples collected from one female and two male subjects who had received a single oral dose of 200 mg TQ. The plasma and blood concentration versus time profiles of TQ are shown in Figure 3A,B, respectively. Urinary TQ and 5,6-OQTQ excretion versus time profiles for the three subjects for four days post dose are shown in Figure 4 and Figure 5, respectively. The data shown in Figure 3A is consistent with other pharmacokinetic studies of TQ such as Brueckner et al. [16] who demonstrated a plasma elimination half-life of 14 days, a Tmax of 12 h and higher concentrations of TQ in blood compared with plasma. The present study extends the pharmacokinetic analysis by measuring urinary TQ and 5,6-OQTQ concentrations. The urine data is consistent with the plasma and blood data where Subject 3 had the highest concentrations of both TQ and 5,6-OQTQ in all three matrixes.

Table 3 shows TQ concentrations in plasma, venous blood and capillary blood from 3 subjects given 200 mg orally administered TQ. The mean +/− (SD) for venous blood/capillary blood is also shown for specific time points. The levels in Finger prick capillary blood and venous blood TQ concentrations were similar, an observation that is in accordance with previous study [8]. This suggests that capillary blood sampling through finger prick rather than venipuncture could be used in larger scale pharmacokinetic studies of TQ in the field.

## 3. Discussion

An UHPLC-MS/MS assay for the rapid quantification of tafenoquine and its major stable metabolite 5,6-OQTQ has been validated in human plasma and urine and for tafenoquine in human blood. Several studies have reported quantification of TQ from plasma and blood [17,18], however the present study is the first to report on the quantification of TQ and the stable metabolite 5,6-OQTQ in plasma and urine. The mass spectrometry of triple quadrupole (QqQ) has high specificity with multiple reaction monitoring (MRM) experiments which help reduce the matrix’s interferences [19]. We also measured TQ and 5,6-OQTQ in venous and capillary blood at several time points and found good agreement between the two blood collections (Table 3). The current gold standard for blood sampling, venous sampling (venepuncture), poses major limitations: it is invasive, requires removal of a relatively large blood volume (>1 mL) and does not allow intensive sampling in vulnerable populations such as children and very ill patients [20]. A recent study of TQ in children showed a good correlation between venous and capillary plasma TQ concentrations [21].

The methods reported here show good accuracy and precision for all three matrices and demonstrate the stability of TQ in blood, plasma and urine. The use of a stable-isotope label form of TQ (SIL_TQ) allowed for the validation of TQ in blood, plasma and urine. This also allowed for the validation of 5,6-OQTQ in plasma and urine. However, matrix effect and variable recovery precluded the validation of 5,6-OQTQ in blood. The matrix effect may be due to heme concentrations in blood that catalyze redox reactions that may be involved in the conversion of TQ (and SIL-TQ) to 5,6-OQTQ. The incorporation of a SIL-form of 5,6-OQTQ would presumably nullify this effect and allow for the validation of 5,6-OQTQ in all three matrices, however a SIL-form of 5,6-OQTQ was not available. The blood 5,6-OQTQ concentrations were measured but at low concentrations (LQC) was outside the acceptance criteria for matrix effects [14,15].

Blood and plasma samples were analyzed on a Qtrap instrument (Sciex 4000 Qtrap) in positive ion MRM mode, while urine samples were analyzed on an orbitrap system (QExactive Plus, Thermo Scientific, Waltham, MA, USA) in positive PRM and full MS modes to allow for additional putative metabolite identification. There are several similarities between the present study and a similar study that validated and examined PQ and 5,6-OQPQ concentrations in plasma and urine in a pharmacokinetic study [12]. The fragmentation pattern of TQ and 5,6-OQTQ as shown in Figure 1 is similar to PQ and 5,6-OQPQ with the alkyl chain of the 8-aminoquinoline undergoing major fragmentation in both studies. The present study found low plasma 5,6-OQTQ concentrations near the LLOQ (1 ng/mL, Appendix A, see the Appendix A) in three subjects given a single oral dose of 200 mg TQ, however their urinary 5,6-OQTQ concentrations were all substantially higher, with a range of 161 to 509 ng/mL for the 24 h collection period at 96 h post TQ dose. These concentrations are also substantially higher than the TQ concentrations in the same samples, which ranged from 9.9 to 232 ng/mL (Appendix A).

The three subjects in the present study who received 200 mg TQ eight days after inoculation with blood-stage *P. falciparum* recrudesced (Table 4) at times that correlate with their urine TQ concentration (Figure 4) and their blood and plasma TQ concentrations (Table 3). Larger pharmacokinetic studies are required to investigate the relationship between TQ and/or 5,6-OQTQ urinary concentrations and malaria treatment outcome, prophylaxis failure and radical cure. Urine monitoring could potentially allow for a non-invasive means of identifying individuals who would require alternate prophylaxis or rescue drugs to avert disease recurrence.

The blood TQ concentrations were on average 1.2-fold higher than corresponding plasma concentrations (Appendix A). This is less than previous TQ pharmacokinetic studies that had mean blood to plasma concentration ratios of 1.7 [8] and 1.8 [9]. It is uncertain why the ratio is lower in the present study but it may in part be explained by lengthy liquid/liquid extraction procedures and chemical additives such as zinc sulfate added during extraction procedures in those studies. Results from the current study show strong similarities to the first-in human TQ pharmacokinetic study with linear kinetics, a Cmax close to 12 h, and an elimination half-life of 14 days [9]. Although there are only three subjects reported here it is interesting to note that all three recrudesced following the 200 mg TQ oral dose with recrudescence times (Table 4) proportional to urinary excretion levels for TQ (Figure 4). The proportionality of urine TQ excretion with treatment outcome may provide a means for clinicians to non-invasively predict drug efficacy in advance of symptom recurrence.

The current study is the first to report on the quantification of TQ and 5,6-OQTQ in urine and its application in a clinical study. Interestingly, urine samples from the study subjects appear to have two variants of 5,6-OQTQ as there are two chromatographic peaks identified as the 304.17/219.08 transition by high resolution accurate mass (orbitrap) mass spectrometry, with the predominant peak corresponding in retention time to the A racemic TQ used to construct the calibration standards and quality control samples. The additional minor peak, seen eluting at 1.96 min could not be eliminated by varying chromatographic conditions, time, or buffers, and was not included in quantification due to retention time difference from the single peak seen in pure racemic 5,6-OQTQ calibration standards and quality control samples. One possibility is that a small amount of 5,6-OQTQ is tightly bound to TQ thus explaining the secondary minor peak elution time corresponding with TQ. Avula et al. [22] reported methods to separate the enantiomers of PQ and carboxy-PQ where both (+) and (−) enantiomers were identified using LC-MS/MS and eluted at similar but not identical times. It has been postulated that retention and enantio-selectivity can depend on hydrophobic interaction, hydrogen binding, ionic bonding and ion pairing interactions [23]. The identity of the additional but minor 5,6-OQTQ peak seen in the urine but not plasma samples from the clinical trial participants, is beyond the scope of the current investigation.

Several clinical studies have correlated CYP2D6 variants with PQ failure and *P. vivax* relapse [11,24]. The orally administered cough medicine dextromethorphan requires CYP2D6 for metabolic conversion [25] and this has been shown to be a useful surrogate for PQ metabolism in a *P. vivax* treatment study [26]. In a recent CYP2D6 genomic study from a *P. vivax*–endemic region of Cambodia it was found that over half (52%) of the participants (*n* = 96) were intermediate metabolizers [24]. In the present study, CYP2D6 phenotype was determined for the three human subjects and all three were found to have normal CYP2D6 metabolism. Thus, the variation in pharmacokinetics (PK) and pharmacodynamics seen between the three subjects in the present study does not appear to be due to variations in CYP2D6 activity. This is consistent with the study by St Jean et al. [27], where they showed that CYP2D6 intermediate metabolizers did not demonstrate a reduced *P. vivax* clearance efficacy when administered TQ compared with PQ. They also demonstrated that hepatic in vitro systems produce only very small amounts of TQ metabolites, not including the 5,6-OQTQ. Thus, the link between TQ metabolism in liver, blood and urine with *P. vivax* radical cure remains elusive.

## 4. Materials and Methods

### 4.1. Chemicals

Tafenoquine succinate (WR238605, N4-(2,6-dimethoxy-4-methyl-5-(3-(trifluoromethyl)phenoxy)quinolin-8-yl)pentane-1, 4-diamine, C_24_H_28_F_3_N_3_0_3_, FW succinate salt 581.59, free base 463.50) and 5,6-orthoquinone TQ succinate (5,6-OQTQ, 8-((5-aminopentan-2-yl)amino)-2-methoxy-4-methylquinoline-5,6-dione, C_16_H_21_N_3_O_3_, FW succinate salt 450.97, free base 303.36) were obtained from WRAIR Inventory Laboratory, Silver Spring, MD, USA. Stable isotope label form of TQ (SIL-TQ, C_24_H_24_(^2^H_4_)F_3_N_2_(^15^N)O_3_.C_4_H_6_O_4_, FW 586.594, free base 468.46) was from Glaxo Smith Kline Research and Development, Isotope Chemistry, Brentford, Middlesex, UK. Formic acid, acetonitrile and methanol were from Thermo Fisher Scientific. Water was purified in a Milli-Q system (Millipore, Bedford, MA, USA).

### 4.2. Instrumentation and Chromatographic Conditions

#### 4.2.1. Analysis of TQ and 5,6-OQTQ in Blood and Plasma

The UHPLC-MS/MS system used for blood and plasma samples was comprised of a Shimadzu UHPLC with a vacuum degasser, refrigerated autosampler, binary pumps and a temperature controlled column oven that was connected to the turbo V source of a Sciex 4000 Qtrap triple quadrupole mass spectrometer operating in MRM mode. The LC column was Waters Atlantis T3 50 mm × 2.1 mm I.D. 3 μm, fitted with an Atlantis Silica T3 VanGuard Cartridge, 100 Å, 3 µm, 2.1 mm × 5 mm (Waters, Milford, MA). The injection volume was 5 µL. The LC conditions consisted of a four minute gradient of 0.1% formic acid (Mobile Phase A) and acetonitrile (Mobile Phase B) as follows: 0.3 min, 10% B; 0.8 min, 70% B; 1.6 min, 70% B; 1.8 min, 10% B. The LC column oven was set to 40 °C, the injector rinse solution was 50% acetonitrile and the autosampler temperature was set to 8 °C. The MS source conditions are detailed below in Table 5.

The 4000 Qtrap mass transitions and voltages are detailed below in Table 6.

The source conditions in Table 5 were determined experimentally to give the sensitivity required for the assays while the product ions (Q3) and voltages in Table 6 were determined by the Compound Optimisation function of the Sciex Analyst software (Version 1.6.2, AB Sciex LLC, Framingham, MA, USA).

#### 4.2.2. Analysis of TQ and 5,6-OQTQ in Urine

The system used for the urine validation and analysis was a Thermo Scientific Vanquish UHPLC with a refrigerated (4 °C) autosampler, binary pumps and a temperature controlled column oven that was connected to the API source of a Thermo Scientific QExactive Plus orbitrap mass spectrometer operating in PRM mode. The urine was analyzed on the high resolution system to enable the identification of additional TQ metabolites (not included in the present study). The LC column was also Waters Atlantis T3 50 mm × 2.1 mm I.D. 3 μm, fitted with an Atlantis Silica T3 VanGuard Cartridge, 100 Å, 3 µm, 2.1 mm × 5 mm (Waters, Milford, MA). The LC gradient was identical to that stated above for the 4000 Qtrap with the exception that methanol containing 0.1% formic acid was used in place of acetonitrile as Mobile Phase B. The QExactive Plus mass spectrometer was operated in positive PRM mode at 35,000 resolution, with Automatic Gain Control at 2 × 10^5^, maximum injection time of 50 ms and an isolation window of 3 *m*/*z*. Inclusion list settings for the PRM experiments are detailed in Table 7 below. Quantification of TQ and 5,6-OQTQ was performed using Tracefinder software (Version 4.1 ThermoFisher, Waltham, MA, USA).

### 4.3. Standard Stock Solutions Preparation

Stock solutions of TQ, 5,6-OQTQ and SIL-TQ were prepared separately at 1 mg/mL in methanol and stored at −80 °C in the dark. Working standard solutions were prepared from the primary stocks at 100, 10 and 1 µg/mL in 50% methanol. Drug-free blood and plasma samples were spiked to 1200, 500, 100, 20, 4 and 1 ng/mL for TQ, and to 200, 80, 20, 5, 2 and 1 ng/mL for 5,6-OQTQ. Quality control samples for blood and plasma were prepared from separate weighings of stock solutions. Quality control samples were spiked to 600, 50 and 2.5 ng/mL for TQ, and to 100, 15 and 2.5 ng/mL for 5,6-OQTQ. For urine analysis, drug-free urines were spiked to 1000, 750, 350, 120, 40 and 10 ng/mL for both TQ and 5,6-OQTQ. Urine quality control samples were spiked to 800, 150 and 20 ng/mL for both TQ and 5,6-OQTQ. Aliquots were stored at −80 °C in the dark.

### 4.4. Method Validation

The UHPLC-MS/MS methods were validated according to the FDA and EMA guidelines [14,15] Selectivity and specificity were determined from six separate blood and plasma and five urine samples to demonstrate that blank and zero calibrators were free of any interferences at the retention times of the analytes and the IS.

For linearity and sensitivity, the calibration curves from six concentrations (1–1200 ng/mL TQ and 1 to 200 ng/mL for 5,6-OQTQ for blood and plasma, and 10–1000 ng/mL for both TQ and 5,6-OQTQ in urine) were analyzed in triplicate and constructed by plotting the peak area ratio of the analyte and the IS against nominal concentration to demonstrate the linearity of the method. The square of the correlation coefficient (r^2^) of the calibration curves for all quantified analytes should be >0.99. The limit of quantification (LOQ) was defined as the concentration with at least a 10:1 signal-to-noise ratio and with precision and accuracy ±20%.

Matrix effect was determined from the responses of TQ and 5,6-OQTQ QCs spiked into six different blood and plasma samples and five different urine samples. The matrix effects were determined from [response of QCs spiked in post-extraction blank samples 100]/[response of the QCs in pure solutions]. The matrix effect should be within ±20%. The % coefficient of variation (%CV) on matrix effects should not be greater than ±15%.

The absolute recovery was determined from [response of extracted QCs in blank samples × 100]/[response QCs spiked in post-extraction blank samples]. The extent of the recovery of analytes and of the IS should be consistent and reproducible. The %CV on recovery should not be greater than ±15%.

Accuracy and precision were determined from the intra-day and inter-day runs. The accuracy should be within ±15% of the nominal concentration at LQC, MQC, and HQC, but ±20% at LLOQ; and the precision (%CV) should be within ±15% at LQC, MQC, and HQC, but ±20% at LLOQ. The limits of detection for TQ and 5,6-OQTQ was not determined as there was no need to detect traces of the analytes below the LOQ.

The stability of TQ and 5,6-OQTQ in the QC standard solutions and samples was only determined under specific conditions during analysis. Short-term stability was determined in an autosampler (8 °C for 24 h) and benchtop/room temperature (25 °C for 4 h). Long-term stability of TQ and 5,6-OQTQ were assessed in the QC standard solutions and samples that were kept at −80 °C to determine an optimal storage condition and total analysis time. The obtained results were compared with the nominal concentrations of the analyte, and it was considered stable if the observed concentrations were within ±15% of the nominal concentration.

### 4.5. Application of the Method

#### 4.5.1. Subjects and Sample Collection

Samples were obtained from three healthy subjects who were administered TQ as part of a phase 1b study to evaluate the blood stage antimalarial activity of a single oral dose (200 mg base) of TQ in healthy subjects eight days after being experimentally infected with *P. falciparum* (~2800 viable *P. falciparum* parasite-infected human erythrocytes on Day 0) [13]. Informed consent was obtained from all subjects involved in the study. Venous blood samples were collected for PK analysis in EDTA anticoagulant blood tubes at 0, 4, 8, 12, 24, 48, 96, 168, 336, 346, 504, 672, 840 and 1032 h after dosing. Plasma was separated by centrifugation at 3000× *g* for 10 min. Capillary blood was also collected by finger prick at 12, 24, 96 and 168 h after dosing. The Becton-Dickinson (BD) contact-activated lancets was used for finger prick capillary blood collections to minimize pain and inconvenience to the participant. The finger prick capillary blood was transferred to BD microtainers containing EDTA for subsequent TQ concentration measurement. Blood and plasma samples were immediately aliquoted and stored at −80 °C. Urines were collected and pooled for each day for the first four days after TQ administration. Daily total void volumes were combined, volume measured and representative aliquots frozen at −80 °C for UHPLC-MS/MS analysis and storage stability assessment.

#### 4.5.2. Sample Preparation

Venous or finger prick capillary blood, or plasma samples (50 µL) were mixed with 200 µL acetonitrile containing 2 ng/mL internal standard (SIL-TQ), vortex mixed for 1 min and centrifuged at 20,000× *g* for 5 min at 4 °C. The supernatant (100 µL) was transferred to a polypropylene 96-well assay plate containing 100 µL 0.1% formic acid, and briefly mixed prior to sealing the plate with a silicon sealing mat. Chromatography for blood and plasma consisted of 4 min gradient runs from 30% to 70% acetonitrile with 0.1% formic acid at 0.5 mL per minute flow rate. The 5,6-OQTQ eluted at 1.6 min while TQ and SIL-TQ eluted at 2.0 min.

Urine samples (100 µL) were mixed with 200 µL acetonitrile containing 2 ng/mL internal standard (SIL-TQ), vortex mixed for 1 min and centrifuged at 20,000× *g* for 5 min. The supernatant (200 µL) was transferred to a polypropylene 96-well assay plate which was sealed with a silicon sealing mat. Injection volume was 5 µL. Chromatography for urine samples consisted of 4 min gradient runs from 30% to 70% methanol with 0.1% formic acid at 0.5 mL per minute flow rate. The 5,6-OQTQ eluted at 2.0 min while TQ and SIL-TQ eluted at 2.3 min.

#### 4.5.3. Pharmacokinetic Parameters

The blood and plasma pharmacokinetic parameters (maximum concentration [Cmax], time to Cmax [Tmax], the area under the curve [AUC], and the area under the curve with extrapolation to infinity [AUC_0-inf_] for TQ and 5,6-OQTQ were determined from the plasma concentration versus time data using PK Solutions V2.0 (Summit Research Services, CO). Urine pharmacokinetic parameters Cmax, Tmax, the amount of drug excreted (AE) and the cumulative amount of drug excreted (CAE) were determined or calculated from the urine AE versus time data. The AE was calculated by the summation of drug excreted (urine concentration (ng/mL) multiplied by urine volume (mL) during the urinary 24 h collection period. CAE was calculated by the accumulation amount of drug excreted after each collection interval.

## 5. Conclusions

Robust methods have been developed to determine TQ and 5,6-OQTQ in human blood, plasma and urine. These methods have been applied to a clinical pharmacokinetic study and found to be suitable. The similarity between venous blood and finger prick capillary blood TQ concentrations from the same individual paves the way for larger studies using finger prick rather than venipuncture sampling for monitoring drug exposure, including compliance. Larger studies are also indicated to clarify the utility of urinary TQ and 5,6-OQTQ concentrations as potential markers of clinical efficacy.

## Figures and Tables

**Figure 1 molecules-27-08186-f001:**
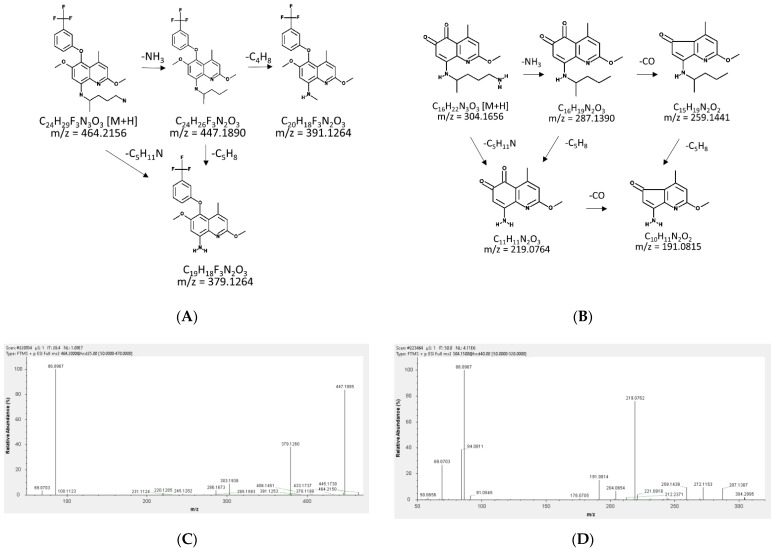
The predicted fragmentation patterns and the MS/MS spectra of tafenoquine (TQ) and 5,6-orthoquinone tafenoquine (5,6-OQTQ). The predicted fragmentation patterns of (**A**) TQ; and (**B**) 5,6-OQTQ. The MS/MS spectra of (**C**) TQ showed the key fragments at *m/z* 379.13 (predicted 379.1265, observed 379.1260, mass error −1.0551), *m/z* 447.19 (predicted 447.1890, observed 447.1885, mass error −1.1181), and m/z 391.13 (predicted 39.1264, observed 391.1253, mass error −2.8124); and (**D**) 5,6-OQTQ showed the key fragments at *m*/*z* 287.14 (predicted 287.1390, observed 287.1387, mass error −1.0448), m/z 259.14 (predicted 259.1441, observed 259.1439, mass error −0.7718) and *m*/*z* 219.08 (predicted 219.0764, observed 219.0762, mass error −0.9129). These MS/MS scans were from the FTMS (Fourier transform mass spectrometer) QExactive Plus (Thermo Fisher, Waltham, MA, USA) positive mode scans at 70,000 resolution.

**Figure 2 molecules-27-08186-f002:**
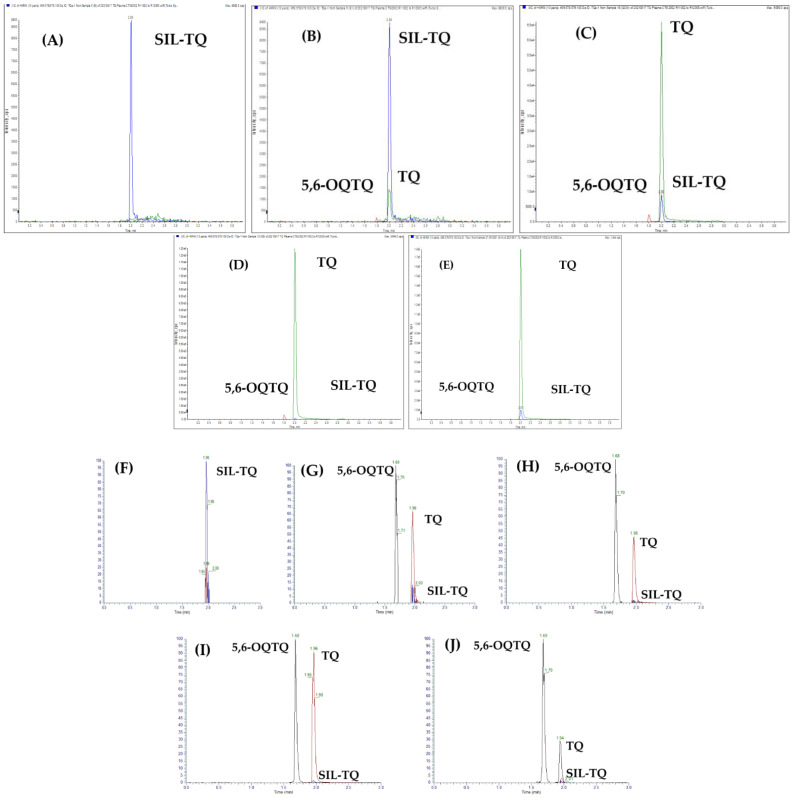
Chromatograms of 5,6-OQTQ, SIL-TQ and TQ in human plasma and urine. (**A**) Blank plasma with internal standard (SIL-TQ, blue peak). (**B**) Spiked plasma at low limit of quantification (LLOQ) for TQ (green peak) and 5,6-OQTQ (red peak). (**C**) Spiked plasma at 50 ng/mL TQ and 15 ng/mL 5,6-OQTQ. (**D**) Spiked plasma at 1200 ng/mL TQ and 200 ng/mL 5,6-OQTQ. (**E**) A clinical trial participant plasma sample. (**F**) Blank urine with internal standard (SIL-TQ). (**G**) Spiked urine at LLOQ for TQ and 5,6-OQTQ. (**H**) Spiked urine at 150 ng/mL TQ and 5,6-OQTQ. (**I**) Spiked urine at 1000 ng/mL TQ and 5,6-OQTQ. (**J**) A clinical trial participant urine sample.

**Figure 3 molecules-27-08186-f003:**
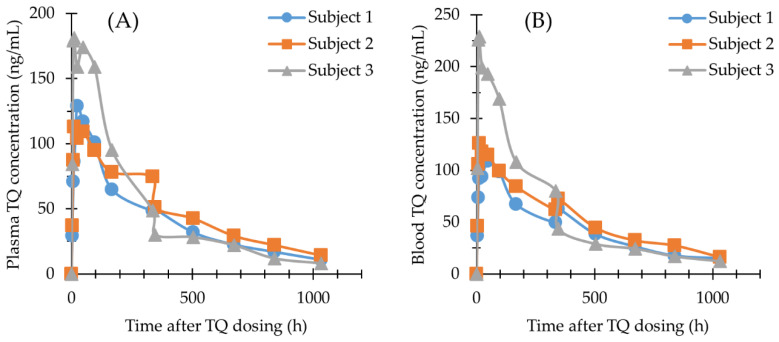
TQ concentration—time profiles in human plasma (**A**) and human blood (**B**) from three subjects given 200 mg TQ orally; Subject 1, (circle line); Subject 2, (square line); Subject 3, (triangle line).

**Figure 4 molecules-27-08186-f004:**
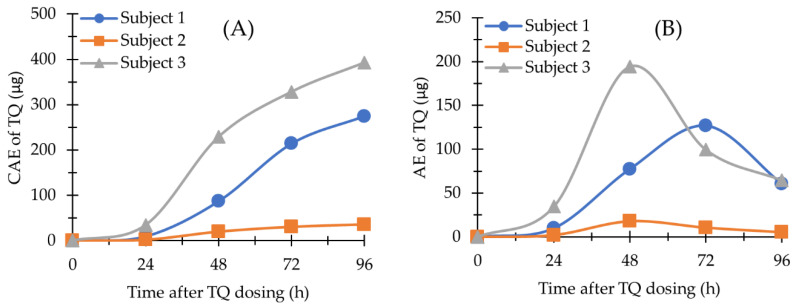
TQ in human urine (*n* = 3). (**A**) The amount of TQ excreted (AE) versus time profile; (**B**) The cumulative amount of TQ excreted (CAE) versus time profile. Time was the endpoint of each 24 h urinary collection period.

**Figure 5 molecules-27-08186-f005:**
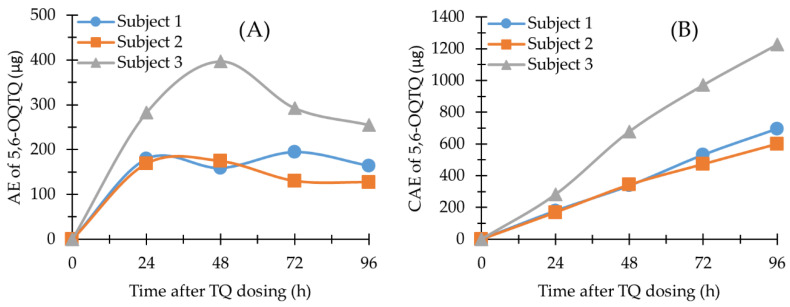
5,6-OQTQ in human urine (*n* = 3). (**A**) AE versus time profile; (**B**) CAE versus time profile. Time was the endpoint of each 24 h urinary collection period.

**Table 1 molecules-27-08186-t001:** Matrix effect and absolute recovery of TQ and 5,6-OQTQ in human blood, plasma and urine.

	Blood		Plasma		Urine	
Analytes	Matrix Effect (%) (Mean ± SD/CV)	Recovery (%) (Mean ± SD/CV)	Matrix Effect (%) (Mean ± SD/CV)	Recovery (%) (Mean ± SD/CV)	Matrix Effect (%) (Mean ± SD/CV)	Recovery (%) (Mean ± SD/CV)
TQ						
LQC	97 ± 14/14.2	187 ± 6/11.9	99 ± 3/2.6	69 ± 5/7.7	87 ± 14/16.0	84 ± 10/11.3
MQC	-	106 ± 6/5.7	-	80 ± 6/7.2	-	85 ± 8.7/10.3
HQC	99 ± 3/3.5	116 ± 4/3.1	98 ± 1/1.0	92 ± 4/3.9	94 ± 3.5/3.7	96 ± 3.5/3.7
5,6-OQTQ						
LQC	64 ± 9/13.5 *	61 ± 16/25.8 **	98 ± 4/3.9	83 ± 7/7.8	86 ± 13/15.5	98 ± 10/10.2
MQC		83 ± 8/10.1	-	110 ± 4/4.1	-	91 ± 10/10.5
HQC	89 ± 5/5.5	82 ± 3.2/3.9	101 ± 7/7.0	94 ± 3/3.2	104 ± 19/17.9	101 ± 3/2.9

LQC; low quality control, MQC; mid quality control, HQC; high quality control, SD; standard deviation, CV; coefficient of variation, (CV = [SD × 100]/mean), -; not determined, *; 5,6-OQTQ matrix effect at LQC > 20%, **; 5,6-OQTQ LQC CV is >15%.

**Table 2 molecules-27-08186-t002:** Accuracy and precision analysis of TQ and 5,6-OQTQ in human plasma, blood and urine.

	Human Plasma		Human Blood		Human Urine	
Analytes	Intra-Day (*n* = 6) Accuracy/Precision (% Mean ± SD %, CV)	Inter-Day (*n =* 14) Accuracy/Precision (% Mean ± SD %, CV)	Intra-Day (*n =* 6) Accuracy/Precision (% Mean ± SD %, CV)	Inter-Day (*n =* 14) Accuracy/Precision (% Mean ± SD %, CV)	Intra-Day (*n* = 5) Accuracy/Precision (% mean ± SD %, CV)	Inter-Day (*n =* 14) Accuracy/Precision (% mean ± SD %, CV)
TQ						
LLOQ	100.5 ± 0.55/0.54	102.3 ± 8.35/8.16	99.82 ± 0.33/0.33	98.40 ± 9.69/9.85	103.00 ± 4.95/4.80	99.74 ± 8.15/8.17
LQC	98.90 ± 5.76/5.91	109.19 ± 6.34/5.85	100.43 ± 9.07/8.94	102.16 ± 8.61/8.43	107.35 ± 6.51/6.06	103.57 ± 5.88/5.68
MQC	98.35 ± 5.44/5.53	101.11 ± 2.93/2.89	102.62 ± 5.68/5.44	97.54 ± 4.78/4.81	104.71 ± 5.51/5.26	103.34 ± 6.98/6.76
HQC	103.35 ± 6.16/5.80	105.60 ± 4.65/4.42	94.42 ± 3.19/3.41	94.09 ± 3.98/4.22	104.74 ± 8.24/7.86	107.61 ± 9.15/8.50
5,6-OQTQ						
LLOQ	101.42 ± 4.27/4.21	98.68 ± 6.96/7.05	-	-	104.58 ± 6.01/5.74	99.81 ± 6.36/6.38
LQC	92.93 ± 6.04/6.58	107.87 ± 8.71/8.06	-	-	103.39 ± 8.00/7.74	100.64 ± 7.83/7.78
MQC	103.05 ± 6.94/6.57	105.14 ± 6.32/6.12	-	-	104.05 ± 5.78/5.55	103.43 ± 12.17/11.77
HQC	101.35 ± 5.04/4.97	104.51 ± 6.35/6.08	-	-	99.44 ± 2.82/2.83	102.58 ± 14.79/14.42

**Table 3 molecules-27-08186-t003:** Plasma, venous blood and finger prick capillary blood concentrations of TQ from subjects given an oral dose of 200 mg TQ (*n* = 3). Capillary blood samples were taken at 12, 24, 96 and 168 h after dosing. Venous/Capillary blood mean values with SD also shown. SD, Standard Deviation; LLOQ, Lower Limit of Quantification (1 ng/mL); N/D, No Data.

	Plasma TQ (ng/mL)			Venous Blood TQ (ng/mL)			Finger Prick Capillary Blood TQ (ng/mL)			Venous/Capillary Blood Mean (±SD) (ng/mL)		
Time (h)	Subject 1	Subject 2	Subject 3	Subject 1	Subject 2	Subject 3	Subject 1	Subject 2	Subject 3	Subject 1	Subject 2	Subject 3
0	BLQ	BLQ	BLQ	BLQ	BLQ	BLQ	N/D	N/D	N/D	N/D	N/D	N/D
4	29.2	37.2	84.2	36.6	46.0	102	N/D	N/D	N/D	N/D	N/D	N/D
8	70.9	87.4	179	73.5	106	226	N/D	N/D	N/D	N/D	N/D	N/D
12	86.2	113	181	92.0	126	229	108	117	179	100 ± 11.3	122 ± 6.4	204 ± 35
24	129	104	159	93.9	118	199	112	124	199	103 ± 12.8	121 ± 4.2	199 ± 0.0
48	117	109	174	109	115	193	N/D	N/D	N/D	N/D	N/D	N/D
96	101	94.9	159	98.6	99.5	169	92.6	97.8	155	96 ± 4.2	99 ± 1.2	162 ± 9.9
168	64.7	78.1	95.2	67.0	84.3	108	72.8	64.7	75.0	70 ± 4.1	75 ± 13.9	92 ± 23.3
336	48.2	74.6	48.6	49.7	61.9	80.1	N/D	N/D	N/D	N/D	N/D	N/D
346	49.7	51.1	30.0	62.6	72.5	43.7	N/D	N/D	N/D	N/D	N/D	N/D
504	31.9	42.6	28.2	38.3	44.6	28.9	N/D	N/D	N/D	N/D	N/D	N/D
672	22.5	29.2	22.1	26.4	32.3	24.3	N/D	N/D	N/D	N/D	N/D	N/D
840	16.9	22.1	11.9	17.6	27.4	16.9	N/D	N/D	N/D	N/D	N/D	N/D
1032	10.9	14.2	8.2	15.0	16.1	12.2	N/D	N/D	N/D	N/D	N/D	N/D

**Table 4 molecules-27-08186-t004:** Pharmacokinetic properties of TQ in human plasma (*n* = 3).

Subject	BW (kg)	Sex	BMI	Cmax (ng/mL)	Tmax (h)	AUC (ng/mL∗h)	AUC_0-inf_ (ng/mL∗h)	t_1/2_ (h)	Recrudescence (day)
1	95.0	Male	29.7	129	24	42,981	48,191	303.3	14
2	87.4	Male	28.8	113	12	50,387	57,902	300.8	9
3	60.6	Female	25.7	181	12	50,675	54,902	357.2	23

BW; body weight; BMI; body mass index (BMI = weight(kg)/[height(m)^2^], Cmax; maximum concentration, Tmax; time to Cmax, AUC; area under the curve, ∗ = multiplied by; AUC_0-inf_; area under the curve with extrapolation to infinity, t_1/2_; elimination half-life, Recrudescence; the time in days after oral dosing that the parasite load increased past a predefined threshold for clearance followed by treatment with artemether/lumefantrine.

**Table 5 molecules-27-08186-t005:** 4000 Qtrap MS source conditions.

Curtain gas	40
CAD gas	Medium
IS (V)	5500
Temp (°C)	600
GS1	50
GS2	50
IHE	On

**Table 6 molecules-27-08186-t006:** Mass transitions and voltages used for the analysis of TQ and 5,6-OQTQ in blood and plasma on the 4000 Qtrap.

Analyte	Q1 (Da)	Q3 (Da)	Time (msec)	DP (v)	EP (v)	CE (v)	CXP (v)
TQ	464.124	447.000	150	76	10	25	12
TQm	304.150	219.100	150	66	10	27	20
TQa	469.252	378.900	150	46	10	31	28

**Table 7 molecules-27-08186-t007:** QExactive Plus Inclusion List for Parallel Reaction Monitoring (PRM).

Mass	Formula	Species	Polarity	CE (V)	Identifier
464.21555	C24H28F3N3O3	+ H	Positive	25	TQ
469.25468	C24H33F3N3O3	+ H	Positive	30	SIL-TQ
304.16557	C16H21N3O3	+ H	Positive	25	5,6-OQTQ

## Data Availability

Not available.

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
