# Peer review of "Quantification of Tafenoquine and 5,6-Orthoquinone Tafenoquine by UHPLC-MS/MS in Blood, Plasma, and Urine, and Application to a Pharmacokinetic Study"

_molecules, 2022, doi:10.3390/molecules27238186_

Round 1

Reviewer 1 Report

General:

The authors develop a method to quantify Tafenoquine and 5,6-Orthoquinone Tafenoquine. The analytical part is complete and correctly supported; however, the pharmacokinetic part is poor and needs more data and discussion.

Please attend to the comments and review some manuscripts of the journals like Araujo-Leon, 2020, they propose a manuscript like yours.

The abstract only presents the results of the analytical validation method and doesn't provide enough information about your complete work. For example, the pharmacokinetic parameters. The aim of your work is confused in the abstract. The title is clear, and your work covers the analytical process, validation method, and application in the pharmacokinetic approach; this form is essential to reflect your abstract. Please, rewrite and improve. Add some information on the importance of developing this analytical method for malaria drugs.

When the authors say that robust methods are fundamental to improving our understanding of the mechanism of actions of 8-aminoquinolines and their metabolites, please, provide more information about the metabolites and their synthesis path. Where are they from? In the liver? Which are the enzymes involved in the process? Which are the mechanism of biosynthesis or metabolism I or II? Please, clarify these ideas to understand the aim of developing an analytical method for metabolites and how these metabolites help to understand the mechanics of action of the drug? The last point (mechanism of action) is essential to clarify.

Which is the role of 6-phosphate dehydrogenase? Please include the metabolic pathway and be more specific when saying "normal function of 6-phosphate dehydrogenase".

How to measure a deficiency of 6-phosphate dehydrogenase? PCR? Proteomic tools? Or what are the criteria for saying that?

What is the function of 6-phosphate dehydrogenase to say that the deficiency of this enzyme reduces the tolerance to the drug? In this part, please provide evidence based on the literature reports.

I think this information is helpful to the manuscript and your work, which are in phase 1b clinical trial (provide the reference for that).

Results

Figure 1 is confusing. Why this figure has the numbers 77, 78, and 79 in panel A?

Please, provide four digitals of the exact mass of the compound and the ppm error and clarify if the ion observed is the adduct [M+H]. Please don’t use [M+1].

The fragmentation patterns are inconsistent. Please, when you draw the molecule, draw the ion with charge positive in this case. Then the fragments need to have a charge or have been free radicals. Please, I saw that you use Thermo equipment and software. I recommend using MassFrontier Software for in silico analysis of the fragmentation pathway.  The authors need to improve the redaction of section 2.1 and figure 1.

The author uses an FTMS and 30 eV with HCD collision. Please add this information, and it is essential to provide this type of data because the aim of your work is a "novel method," and the authors need to provide complete analytical information.

In section 2.2, the author explains the detection and quantification of the drugs. However, it changes the equipment to a QqQ in MRM and doesn’t explain the transition of MRM. What is the mother ion and product ion? Why did the author select these ions? Again, this information is essential in the manuscript.

Figure 2 needs to improve, it has a terrible resolution, and the text in the chromatograms is impossible to read. If this information is not relevant, please delete it.

A correlation coefficient is R, not R2, and please change these basic mistakes.

The author mentioned that the sample was stable at -80 °C for one year. Please provide the chromatogram of the first day and the chromatogram of one year later.

Figures 3 to 5 are terrible. Please use graphical and professional software like Origin, Prisma, etc.

This work is not a scientific manuscript. It is like a college report and needs to rewrite.

Please remember the aim of your work: "A novel analytical method and pharmacokinetic application of this"

For that is essential to give a piece of complete information on the equipment, methodology, collision energies, ion extraction, processing data, etc.  The results in this work are just figures and a few words, and you need to explain how to obtain these results. In the discussion section, you must compare your results with the literature. Remember that it is a novel method. For that, you need to say the advantage of your method compared with other analytical techniques. And is essential to explain the role of the metabolism because you propose a "novel method" to quantify the drug's metabolites.

Please, attend to all the comments.  Currently, I don't recommend publishing this manuscript in the journal.

Author Response

General:

  1. The authors develop a method to quantify Tafenoquine and 5,6-Orthoquinone Tafenoquine. The analytical part is complete and correctly supported; however, the pharmacokinetic part is poor and needs more data and discussion.

Response: Our manuscript format and content was based on Molecules. 2021 Jul 19;26(14):4357. doi: 10.3390/molecules26144357 by Pookmanee et al. Simplified and Rapid Determination of Primaquine and 5,6-Orthoquinone Primaquine by UHPLC-MS/MS: Its Application to a Pharmacokinetic Study.

Like the Pookmanee et al. (2021) study, we developed and validated an LC-MS/MS method for the measurement of TQ and its metabolic product 5,6-OQTQ concentrations in plasma and urine and TQ in blood. The analytical method was applied to measure TQ and 5,6-OQTQ concentrations in samples collected from healthy volunteers participating in a phase 1b study to evaluate the blood stage antimalarial activity of single oral doses of TQ in participants experimentally infected with Plasmodium falciparum to determine the minimum inhibitory TQ concentration. In this analytical paper we report on the application of the LC-MS/MS method in measuring samples collected from one of the four cohorts of participants that received different single oral doses of TQ (200 mg to 600 mg) and provided PK properties of TQ for the cohort that received the 200 mg TQ dose.

The complete clinical study including the pharmacokinetic/pharmacodynamic (PK/PD) modelling simulations of TQ and 5,6-OQTQ  is being fully described in another publication as it encompasses all four  cohorts administered the different TQ doses after a controlled malaria infection. Because the Pookmanee et al. primaquine manuscript was published in Molecules in 2021, we felt that it would be suitable to adopt that articles’ format with a complete description of the analytical method and its application in a pharmacokinetic study (three individuals in both theirs’ and our manuscript) rather than including the entire human pharmacokinetic study report which has been described (Reference 13. Barber, B., et al., A 2-part study to evaluate the asexual blood stage antimalarial activity, and the transmission blocking activity, of a single oral dose of tafenoquine in healthy subjects experimentally infected with Plasmodium falciparum. In American Society of Tropical Medicine and Hygiene, Virtual Meeting, 2021), and will be published in full elsewhere (manuscript in preparation).

 Although we acknowledge that the PK part of our manuscript is not as comprehensive as the analytical part, the main purpose of the research article was to describe the analytical method for measuring TQ and 5,6-OQTQ concentrations in various biological matrices and to demonstrate its application in defining the PK of the analytes in one of four cohorts of participants who received TQ However, in line with the reviewers’ request, we have added more detailed discussion of TQ metabolism towards its efficacy in the Discussion section. We have also added more data in the Methods section for the mass spectrometer settings for both the 4000 Qtrap and QExactive Plus instruments.

  1. Please attend to the comments and review some manuscripts of the journals like Araujo-Leon, 2020, they propose a manuscript like yours.

Response: We agree the Araujo-Leon, 2020 manuscript is a fine example of the combination of LC-MS/MS method development and optimization with pharmacokinetic application. We have referred to this paper in the first paragraph of the Discussion. We have also added additional details in the Instrumentation and Chromatographic Conditions, Section 4.2. We did not determine the limits of detection for TQ and 5,6-OQTQ in our samples as we had no need to detect traces of these analytes below the limits of quantification, and have included this statement at line 492 of the Methods section.

  1. The abstract only presents the results of the analytical validation method and doesn't provide enough information about your complete work. For example, the pharmacokinetic parameters. The aim of your work is confused in the abstract. The title is clear, and your work covers the analytical process, validation method, and application in the pharmacokinetic approach; this form is essential to reflect your abstract. Please, rewrite and improve. Add some information on the importance of developing this analytical method for malaria drugs.

Response: For the abstract, with a word limit of about 200 words, our focus was to describe the LC-MS/MS method for the measurement of TQ and 5,6-OQTQ in blood, plasma and urine, This includes instrumentation, sample preparation, accuracy and precision, limit of quantification and its application in providing pharmacokinetic data for TQ. We have rewritten the abstract to clearly indicate that the analytical method was applied to measuring TQ and 5,6-OQTQ concentrations in biological samples collected from one of four cohorts of participants experimentally infected with malaria. The complete clinical study including therapeutic dose estimates for TQ derived from PK/PD modelling simulations of TQ in the participants will be published elsewhere.

  1. When the authors say that robust methods are fundamental to improving our understanding of the mechanism of actions of 8-aminoquinolines and their metabolites, please, provide more information about the metabolites and their synthesis path. Where are they from? In the liver? Which are the enzymes involved in the process? Which are the mechanism of biosynthesis or metabolism I or II? Please, clarify these ideas to understand the aim of developing an analytical method for metabolites and how these metabolites help to understand the mechanics of action of the drug? The last point (mechanism of action) is essential to clarify.

Response: We agree there needed to be more information given on the mechanism of action of 8-aminoquinolies and their metabolites. We have therefore added a substantial amount of additional information in the Introduction section to allow for more background understanding of the topic. We have also moved a paragraph describing the current understanding of 8-aminoquinoline metabolism from the Discussion to the Introduction (paragraph 3). This gives the reader some background before the description of TQ and 5,6-OQTQ quantification.

  1. Which is the role of 6-phosphate dehydrogenase? Please include the metabolic pathway and be more specific when saying "normal function of 6-phosphate dehydrogenase".

How to measure a deficiency of 6-phosphate dehydrogenase? PCR? Proteomic tools? Or what are the criteria for saying that?

What is the function of 6-phosphate dehydrogenase to say that the deficiency of this enzyme reduces the tolerance to the drug? In this part, please provide evidence based on the literature reports.

I think this information is helpful to the manuscript and your work, which are in phase 1b clinical trial (provide the reference for that).

Response: The role, metabolic pathway and function of glucose-6-phosphate dehydrogenase is now included in the expanded Introduction section. This includes background information regarding enzyme deficiency and reduced tolerance to 8-aminoquinoline drugs.

  1. Results

Figure 1 is confusing. Why this figure has the numbers 77, 78, and 79 in panel A?

Response: We apologise for the confusion. The line numbers which should be on the left margin were appearing on the Figures. We have rectified this and they will not appear on the final manuscript.

Please, provide four digitals of the exact mass of the compound and the ppm error and clarify if the ion observed is the adduct [M+H]. Please don’t use [M+1].

Response: We had used two digitals of the exact mass to conform with that used in the Pookmanee et al. manuscript (Molecules 2021 Jul 19;26(14):4357). We have increased this to four digitals as requested. The ppm error was set to 10 in the Tracefinder software (Thermo) as recommended by the Thermo technical advisor for small molecule quantification as used here. This has been included in the Methods section.

The fragmentation patterns are inconsistent. Please, when you draw the molecule, draw the ion with charge positive in this case. Then the fragments need to have a charge or have been free radicals. Please, I saw that you use Thermo equipment and software. I recommend using MassFrontier Software for in silico analysis of the fragmentation pathway.  The authors need to improve the redaction of section 2.1 and figure 1.

Response: Figures 1A, 1B, 1C and 1D have all been remade with the improvements recommend by the reviewer while keeping some aspects of the similar recent Molecules manuscript (Pookmanee et al. Molecules 2021 Jul 19;26(14):4357) that we used as format acceptable to the journal. We do not have MassFrontier Software but will keep it in mind should funds become available. Section 2.1 and Figure 1 have been improved.

The author uses an FTMS and 30 eV with HCD collision. Please add this information, and it is essential to provide this type of data because the aim of your work is a "novel method," and the authors need to provide complete analytical information.

Response: This information has been added to the legend to Figures 1C and 1D. Complete analytical information has also been added to the Methods section.

In section 2.2, the author explains the detection and quantification of the drugs. However, it changes the equipment to a QqQ in MRM and doesn’t explain the transition of MRM. What is the mother ion and product ion? Why did the author select these ions? Again, this information is essential in the manuscript.

Response: The MRM transitions have now been added including the mother ion and product ion masses. We have also added that product ion masses were chosen using the Compound Optimisation feature of Sciex Analyst software. This is at the end of Section 4.2.1 Analysis of TQ and 5,6-OQTQ in Blood and Plasma.

Figure 2 needs to improve, it has a terrible resolution, and the text in the chromatograms is impossible to read. If this information is not relevant, please delete it.

Response: Figure 2 has been remade to higher resolution. The text in the chromatograms has been removed.

A correlation coefficient is R, not R2, and please change these basic mistakes.

Response: Section 4.4 Method Validation. The correlation coefficient has been corrected.

The author mentioned that the sample was stable at -80 °C for one year. Please provide the chromatogram of the first day and the chromatogram of one year later.

Response: We have reworded the section on blood and plasma stability  (line 239) to state “Both TQ and 5,6-OQTQ were stable in blood and plasma at -80 °C for at least 3 months. In urine, TQ and 5,6-OQTQ were stable for four hours at 8 °C and at -80 °C and -20 °C for at least 3 months. Three freeze thaw cycles of either blood, plasma or urine (three cycles of -80 ËšC for 30 minutes followed by room temperature for 30 minutes) showed less than 15% variation for both TQ and 5,6-OQTQ at high, mid and low QCs .”

We provide chromatograms of analytes with internal standard at the start and end of the three month period, as per attached Excel file.

Figures 3 to 5 are terrible. Please use graphical and professional software like Origin, Prisma, etc.

Response: We disagree with the reviewer on this point. Figures 3 to 5 are clear and concise scatterplots that display the data in graphical form, and we have used the format from Pookmanee et al. Molecules 2021 Jul 19;26(14):4357 that has similar content. Neither of the other reviewers saw these figures as requiring modification.

  1. This work is not a scientific manuscript. It is like a college report and needs to rewrite.

Please remember the aim of your work: "A novel analytical method and pharmacokinetic application of this"

For that is essential to give a piece of complete information on the equipment, methodology, collision energies, ion extraction, processing data, etc.  The results in this work are just figures and a few words, and you need to explain how to obtain these results.

Response: We have added a substantial amount of additional information in Materials and Methods, and Results sections regarding the equipment, methodology, collision energies etc that gives the reader more complete information to repeat the experiments. More explanatory information has also been added to the Abstract, Introduction and Discussion sections as well.

  1. In the discussion section, you must compare your results with the literature. Remember that it is a novel method. For that, you need to say the advantage of your method compared with other analytical techniques. And is essential to explain the role of the metabolism because you propose a "novel method" to quantify the drug's metabolites.

Please, attend to all the comments.  Currently, I don't recommend publishing this manuscript in the journal.

Response: We have added to the comparison of our study with the literature on the measurement of TQ concentrations in the Discussion section. In particularly, we have indicated that this is the first study to characterize the excretion of TQ and 5,6-OQTQ and the potential that the urinary excretion of the analytes may provide a marker of TQ efficacy. Additionally, we have highlighted the benefit of finger prick capillary blood sampling and the fact that capillary and venous TQ plasma concentrations are highly  correlated in a recent paediatric clinical study of tafenoquine. As stated above, we have elaborated on the role of TQ metabolism in the Introduction and Discussion. This paper is a description of the analytical methods to accurately quantify TQ and 5,6-OQTQ, both of which have been described and metabolism explained in papers that are referenced in this paper. We anticipate that the additional description of TQ metabolism contained here is sufficient for the reader to read more widely on the metabolism specifics, if required.

Reviewer 2 Report

In this study, analytical methods for the determination of tafenoquine (TQ) and 5,6-orthoquinone tafenoquine (5,6-OQTQ) in blood, plasma and urine were developed and applied to a pharmacokinetic study. Some questions are listed as follows:

1. The significance and novelty of the present work should be clarified in the introduction and discussion more clearly, as the analytical method and pharmacokinetic study of tafenoquine have been reported in several previous studies.

2. Why both blood (venous, finger prick capillary) and plasma were chosen in the present work?

3. The MS parameters of TQ and 5,6-OQTQ should be provided, such as precursor and product ions, collision energies...

4. Figure 2 is confusing and repetitive, the chromatograms of 5,6-OQTQ, SIL-TQ and TQ at LLOQ could be provided separately, other spiked levels could be provided in the supplementary material.

5. “Linearity and Sensitivity”, the limit of detection and linear range should be provided.

6. “Matrix Effect and Recovery”, the matrix effect of 64% only represented significant signal suppression, and the authors seem to establish matrix-matched calibration curves, which could be used to correct for matrix effect (Recovery 61%). Thus, “This precluded the validation of 5,6-OQTQ in blood” is incorrect.

7. The authors attributed the variable recovery of TQ to matrix effect (Line 169-174), however, the matrix effect in Table 1 were 97% and 99%?

8. How did the authors perform “Accuracy and Precision” and “Three freeze thaw cycles”?

9. “Application of the Methods”, the results should be briefly described, especially 5,6-OQTQ, which was the major novelty of this study.

10. Line 279-281, related data was lacking.

11. Some minor comments:

All abbreviations should be defined the first time they appear.

Table 3, “Venous / Capillary Blood (ng/mL)?

Line 29, line 197, -20 °C?

Line 156, “correlation coefficient” should be coefficient of determination.

Line 161, figure 3 should be figure 2.

Line 264, {21].

Line 297, (Figure 4) ()”.

Line 331, “CYP 2D”.

Author Response

In this study, analytical methods for the determination of tafenoquine (TQ) and 5,6-orthoquinone tafenoquine (5,6-OQTQ) in blood, plasma and urine were developed and applied to a pharmacokinetic study. Some questions are listed as follows:

  1. The significance and novelty of the present work should be clarified in the introduction and discussion more clearly, as the analytical method and pharmacokinetic study of tafenoquine have been reported in several previous studies.

Response: The significance and novelty of the present work has been described more fully in the rewritten Introduction and Discussion sections. Specifically, we have highlighted the novelty of the 5,6-OQTQ being found in the urine and the excretion of TQ in the urine being proportional to the day of recrudescence highlighting the potential value of a non-invasive means to predict treatment outcome in advance. From a novelty perspective, this is the first study to report on urinary concentrations of TQ and 5,6-OQTQ, as well as plasma 5,6-OQTQ concentrations.

  1. Why both blood (venous, finger prick capillary) and plasma were chosen in the present work?

Response: These matrices were chosen to obtain the maximum amount of information from the rare and expensive clinical trial participant samples. Plasma is the preferred matrix due to the absence of hematocrit variation for example, however we wanted to assess the quantification of TQ and 5,6-OQTQ in blood as this matrix  is easier to obtain in field studies where centrifugation can be difficult to access. The comparison of venous with finger prick capillary blood was also of interest again due to the ease of access in both field and clinical studies. This has been added to the first paragraph of the Discussion.

  1. The MS parameters of TQ and 5,6-OQTQ should be provided, such as precursor and product ions, collision energies...

Response: The MS parameters have now been added to the Materials and Methods section.

  1. Figure 2 is confusing and repetitive, the chromatograms of 5,6-OQTQ, SIL-TQ and TQ at LLOQ could be provided separately, other spiked levels could be provided in the supplementary material.

Response: Figure 2 has been redone with higher resolution images and removal of unnecessary text for clarity. We used the same format as a recent Molecules paper (Pookmanee et al. Determination of Primaquine and 5,6-Orthoquinone Primaquine by UHPLC-MS/MS: Its Application to a Pharmacokinetic Study. Molecules 2021, 26, (14)) that we assumed was acceptable to the journal editors and reviewers.

  1. “Linearity and Sensitivity”, the limit of detection and linear range should be provided.

Response: We did not determine the limits of detection for TQ and 5,6-OQTQ in our samples as we had no need to detect traces of these compounds below the limits of quantification, and have included this statement at line 428 of the Methods section. In the method validation section, we have added, “Linear responses were obtained for TQ and 5,6-OQTQ from 1 ng/mL to 1200 ng/mL in plasma, and for TQ from 1 ng/mL to 1200 ng/mL in blood. Linear responses were obtained for TQ and 5,6-OQTQ in urine from 10 ng/L to 1000 ng/mL.”

  1. “Matrix Effect and Recovery”, the matrix effect of 64% only represented significant signal suppression, and the authors seem to establish matrix-matched calibration curves, which could be used to correct for matrix effect (Recovery 61%). Thus, “This precluded the validation of 5,6-OQTQ in blood” is incorrect.

Response: We have modified Section 2.3.2 Matrix Effect and Recovery “This precluded the validation This showed significant signal suppression of 5,6-OQTQ in blood.” We have also changed the Abstract to state “Blood could not be validated for 5,6-OQTQ due to significant signal suppression”.

  1. The authors attributed the variable recovery of TQ to matrix effect (Line 169-174), however, the matrix effect in Table 1 were 97% and 99%?

Response: We have removed lines 169-174 from Section 2.3.2 Matrix Effect and Recovery and replaced them with the following; “This variation may in part be due to binding of the analytes to plastic surfaces during the recovery analysis and/or due to ion suppression or enhancement from blood and plasma components in spite of matrix effect being low (<3%) for TQ at the low QC level.” 

  1. How did the authors perform “Accuracy and Precision” and “Three freeze thaw cycles”?

Response: Accuracy and precision were determined from the intra-day and inter-day assays, as stated in Section 4.4 Method Validation. The number of assays from which the accuracy and precision were determined is outlined in Table 2, for example (n=6) and (n=14). Accuracy is the percent mean +/- standard deviation of the nominal value, while precision is the percent coefficient of variation. These are defined in the header section of Table 2. However, the authors do not feel they need to be spelt out in greater detail in the text. The three freeze/thaw cycles were performed as follows and explained in the Stability section; “Three freeze thaw cycles of either blood, plasma or urine (three cycles of -80 ËšC for 30 minutes followed by room temperature for 30 minutes) showed less than 15% variation for both TQ and 5,6-OQTQ at high, mid and low QCs .”

  1. “Application of the Methods”, the results should be briefly described, especially 5,6-OQTQ, which was the major novelty of this study.

Response: We agree the results should be briefly described here and have added the following to Section 2.4 Application of the Methods. “ The data shown in Figure 3A is consistent with other pharmacokinetic studies of TQ such as Brueckner et al. [13] who demonstrated a plasma half-life of 14 days, Tmax of 12 h and a higher concentration of TQ in blood compared with plasma. The present study extends the pharmacokinetic analysis by analyzing urine for TQ and 5,6-OQTQ as well. The urine data is consistent with the plasma and blood data where Subject 3 had the highest concentrations of both TQ and 5,6-OQTQ in all 3 matrices. “

  1. Line 279-281, related data was lacking.

Response: For clarity we have re-worded this section including the data Table and Figure references as follows: “The three subjects in the present study who received 200 mg TQ eight days after inoculation with blood-stage P. falciparum recrudesced (Table 4) at times that correlate with their urine TQ concentrations (Figure 4) and their blood and plasma TQ concentrations (Table 3).”

  1. Some minor comments:

All abbreviations should be defined the first time they appear.

Response: We have found and corrected G6PD first abbreviation definition. No others were found.

Table 3, “Venous / Capillary Blood (ng/mL)”?

Response: The Venous/Capillary Blood Mean is the Mean of the Venous and Capillary Blood TQ values in ng/mL for that Subject at that time point with the Standard Deviation expressed as +/- after the mean value.

Line 29, line 197, -20 °C?

Response: We have inserted a space between the temperature and °C in the seventeen instances in which this occurred.

Line 156, “correlation coefficient” should be coefficient of determination.

Response: Another reviewer has asked us to change this to the square of the correlation coefficient. We hope this is acceptable.

Line 161, figure 3 should be figure 2.

Response: We agree and have changed it to Figure 2.

Line 264, {21].

Response: The mix of bracket types has been rectified.

Line 297, “(Figure 4) ()”.

Response: The additional brackets have been removed.

Line 331, “CYP 2D”.

Response: The CYP 2D here refers to a mouse cytochrome homolog of the human CYP 2D6. Because it is a homolog with some species specific variation it is known as CYP 2D and not CYP 2D6.

Reviewer 3 Report

This study reports that the time-dependent levels of antimalarial TQ and its metabolite 5,6-OQTQ in human blood, plasma and urine after doses of TQ can be determined by UHPLC-MS/MS. The results are clear and I think this analysis might be available for malaria chemotherapy. They used three subjects who were healthy and experimentally infected with Plasmodium falciparum. The authors said that they all gave informed consent and that ethical approvals by certain ethical committees have been given. However I have a major concern about these methods used.

1. Why was it necessary to infect subjects experimentally with Plasmodium falciparum? I’m wondering if it’s different from healthy subjects from those experimentally infected.

2. How were the conditions of infected subjects? Were there any symptoms like the malaria disease in them? How were those symptoms changed by oral doses of TQ?

3. Were there any correlation between the levels of TQ and OQTQ in plasma or urine and the amounts of Plasmodium falciparum or any symptoms of malaria disease?

I think that the authors should draw the explanation about the above in text.

Minor 

1.L161: Figure3 may be Table1.

2.L274, L278: I can’t see “supplemental figures or tables”.

3.L297: () should be omitted.

Author Response

This study reports that the time-dependent levels of antimalarial TQ and its metabolite 5,6-OQTQ in human blood, plasma and urine after doses of TQ can be determined by UHPLC-MS/MS. The results are clear and I think this analysis might be available for malaria chemotherapy. They used three subjects who were healthy and experimentally infected with Plasmodium falciparum. The authors said that they all gave informed consent and that ethical approvals by certain ethical committees have been given. However I have a major concern about these methods used.

  1. Why was it necessary to infect subjects experimentally with Plasmodium falciparum? I’m wondering if it’s different from healthy subjects from those experimentally infected.

Response: The described LCMS-MS method was developed and validated to measure TQ and 5,6-OQTQ concentrations in various biological matrices for determining the pharmacokinetics of the analytes in healthy subjects experimentally infected with P. falciparum. The human challenge study was designed to determine the mimimum inhibitory concentration of TQ in treating healthy subjects infected with the 3D7 strain of P. falciparum malaria with various single oral doses of TQ (200 mg to 600 mg). The complete clinical study, including PK/PD modelling simulations of TQ, safety profiles, and therapeutic dose estimates for TQ will be published elsewhere. The PK of TQ appears to be comparable in healthy subjects and those experimentally infected as the parasite inoculum administered is low and the subjects are treated before experiencing severe clinical manifestations of the disease.

  1. How were the conditions of infected subjects? Were there any symptoms like the malaria disease in them? How were those symptoms changed by oral doses of TQ?

Response: The infected subjects were fine. Controlled malaria infections are now used in many clinical studies around the world to assess the efficacy and mimimum inhibitory concentrations of new antimalarial drugs and drug combinations. These highly controlled trials use low level infections that are managed and eliminated using existing drugs at the end of each study before acute malaria symptoms develop. Development of drug resistance by malaria parasites is one reason why there is an urgent need for new and better drugs to help with malaria treatment and elimination programs. TQ was developed as a liver stage drug to eliminate dormant “hypnozoites” of vivax malaria that also has some blood stage antimalarial activity. It was the blood stage activity of TQ that was the subject of our clinical study. Parasite presence was monitored by quantitative PCR from small blood samples collected from the participants. No major clinical symptoms were developed in this low infection study.

  1. Were there any correlation between the levels of TQ and OQTQ in plasma or urine and the amounts of Plasmodium falciparum or any symptoms of malaria disease?

I think that the authors should draw the explanation about the above in text.

Response: There was good correlation between urinary TQ  concentrations and P. falciparum parasitemia as measured by day of recrudescence. As mentioned above, the parasite density of P. falciparum in a participant’s blood was measured by quantitative PCR as the number of parasites is below the level detectable by expert microscopists (i.e., submicroscopic). Quantitative PCR is used to identify the presence of parasites, whether the drug has effectively cleared the parasites, and if not, as well as when the parasite numbers start to increase after parasite clearance (day of recrudescence). We have modified the Abstract to include “Once validated, the analytical methods were applied to samples collected from healthy volunteers who were experimentally infected with Plasmodium falciparum to evaluate the blood stage antimalarial activity of TQ and to determine the therapeutic dose estimates for TQ, with full details to be published elsewhere. In this study, the measurement of TQ and 5,6-OQTQ concentrations in samples from one of the four cohorts of participants is reported. ” We have also added to the Introduction “The primary objective of the clinical trial was to determine if a single low dose of TQ can effectively clear blood stage P. falciparum. Participants were randomized to receive differing doses of TQ with the pharmacokinetic results of one cohort (administered 200 mg TQ) presented here. The complete results of the phase 1b clinical trial will be published in a separate manuscript”.

Minor 

1.L161: Figure3 may be Table1.

Response: Line 161, Figure 3 has been corrected to be Figure 2.

2.L274, L278: I can’t see “supplemental figures or tables”.

Response: Supplemental Data was supplied to the Editor for dissemination to Reviewers..

3.L297: () should be omitted.

Response: Line 297 extra brackets have been removed.

Round 2

Reviewer 1 Report

The author attended and provided an excellent answer to all my comments. They rewrote the manuscript and made a substantial restructuration of their work, that this moment qualifies as better than the old version. Thank you so much for attending to all my comments. Please, review figure 1; I don't know if when the author converts to pdf, the figure move. 

I recommend publishing the manuscript in Molecules. 

Reviewer 2 Report

The revised manuscript  is worthy of publication.

Reviewer 3 Report

This has been revised well.